# Enantioselective access to tricyclic tetrahydropyran derivatives by a remote hydrogen bonding mediated intramolecular IEDHDA reaction

Min Jin [1,6], Congyun Tang [2,6], Yingying Li[3], Shuai Yang [1], Ying-Tao Yang[1], Lin Peng [2], Xiao-Nian Li[1], Wenjing Zhang [4✉], Zhili Zuo [1✉], Fabien Gagosz [5] & Liang-Liang Wang [1✉]

Inverse-electron-demand-hetero-Diels-Alder reactions of alkenes with α,β-unsaturated keto compounds allow rapid access to the tetrahydropyran ring found in numerous natural products and bioactive molecules. Despite its synthetic interest, catalytic asymmetric versions of this process remain underdeveloped, especially regarding the use of non-activated alkenes reacting with α,β-unsaturated ketone or aldehyde, for which no report can be found in the literature. Herein, we describe the catalytic inverse-electron-demand-hetero-Diels-Alder reactions between neutral alkenes and an α,β-unsaturated ketones or aldehydes to produce a variety of trans-fused [5,6,8] tricyclic structures containing a central, chiral tetrahydropyran ring. This complex transformation, which is achieved using a chiral phosphoric acid, allows for the formation of four stereogenic centers in a single step with high regio-, diastereo- and enantioselectivity (up to 99% ee). Such level of stereocontrol could be achieved by a key remote double hydrogen atom bonding interaction between the linear substrate and the catalyst.

[1] State Key Laboratory of Phytochemistry and Plant Resources in West China, Kunming Institute of Botany, Chinese Academy of Sciences, 650201 Kunming, PR China. [2] School of Food and Chemical Engineering, Shaoyang University, 422000 Shaoyang, PR China. [3] School of Chemical Engineering, Sichuan University of Science & Engineering, 643000 Zigong, PR China. [4] College of Chemistry and Molecular Engineering, Zhengzhou University, 450001 Zhengzhou, Henan Province, PR China. [5] Department of Chemistry and Biomolecular Sciences, University of Ottawa, K1N 6N5 Ottawa, Canada. [6] These authors contributed equally: Min Jin, Congyun Tang. ✉email: zhangwj@zzu.edu.cn; zuozhili@mail.kib.ac.cn; wangliangliang@maik.kib.ac.cn

Trans-fused [5,6] bicycles containing a tetrahydropyran ring are privileged structural motifs that can be found in numerous natural products and/or bioactive compounds. For example, natural products Englerin A (**1**) and Parvifolal A (**2**) (Fig. 1a), which, respectively, exhibit selective cytotoxic activity for renal cancer cell lines and inhibitory activity against sbLOX-1, feature a central trans-fused tetrahydropyranocyclopentane motif A[1,2]. The analogous nitrogen atom-containing bicyclic structure —i.e. the octahydropyranopyrrole motif B is also present in a series of synthetic bioactive molecules as exemplified by the tachykinin receptor antagonist **3** and the alpha-1 adrenergic antagonist **4** (Fig. 1b)[3,4]. Despite the importance and the wide occurrence of structural motifs A and B, very little attention has been brought to their enantioselective construction. This situation is in sharp contrast with the numerous efforts made to access the analogous full carbon trans-fused [5,6] hydrindane and tetra-hydroindane bicyclic motifs (for selected works involving the synthesis of the challenging *trans*-hydrindane motif see ref. [5] and for selected studies on the synthesis of the *trans*-tetrahydroindane motif see ref. [7])[5–8]. Among the various approaches that could be considered to build chiral tetrahydropyran-contained skeletons, the most direct one would most probably consists of imple-menting an asymmetric inverse-electron-demand hetero-Diels–Alder (IEDHDA) reaction between an alkene and an unsaturated carbonyl electrophile (for selected reviews on asym-metric IEDHDA reactions see ref. [9])[9–12]. Although the inter-molecular asymmetric IEDHDA reaction involving electron-rich olefins and α,β-unsaturated carbonyl electrophiles is well docu-mented, non-activated alkene substrates have been less studied in such type of reaction due to their comparatively low reactivity[9–12].

Previous investigations in this field are summarized in Fig. 2. Cheng and co-workers initially reported the enantioselective inter-molecular Diels–Alder reaction between styrene and β,γ-unsaturated ketone esters catalyzed by a Lewis acid-assisted Brønsted acid com-plex (Fig. 2a)[13]. Later, Ishihara and co-workers disclosed that bis(oxazoline)/copper(II) species could promote the enantioselective

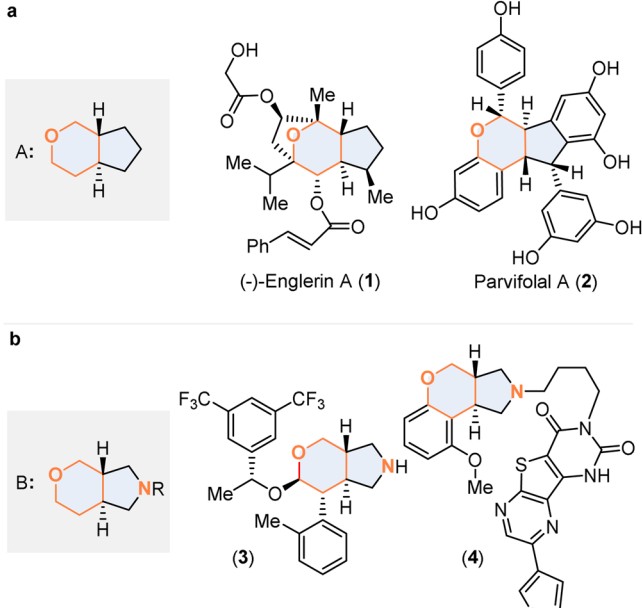

**Fig. 1 Bioactive molecules containing a trans-fused tetrahydropyran bicyclic motif. a** Selected bioactive natural products with a *trans*-fused tetrahydropyranocyclopentane motif. **b** Selected synthetic bioactive molecules with a *trans*-fused octahydropyranopyrrole motif.

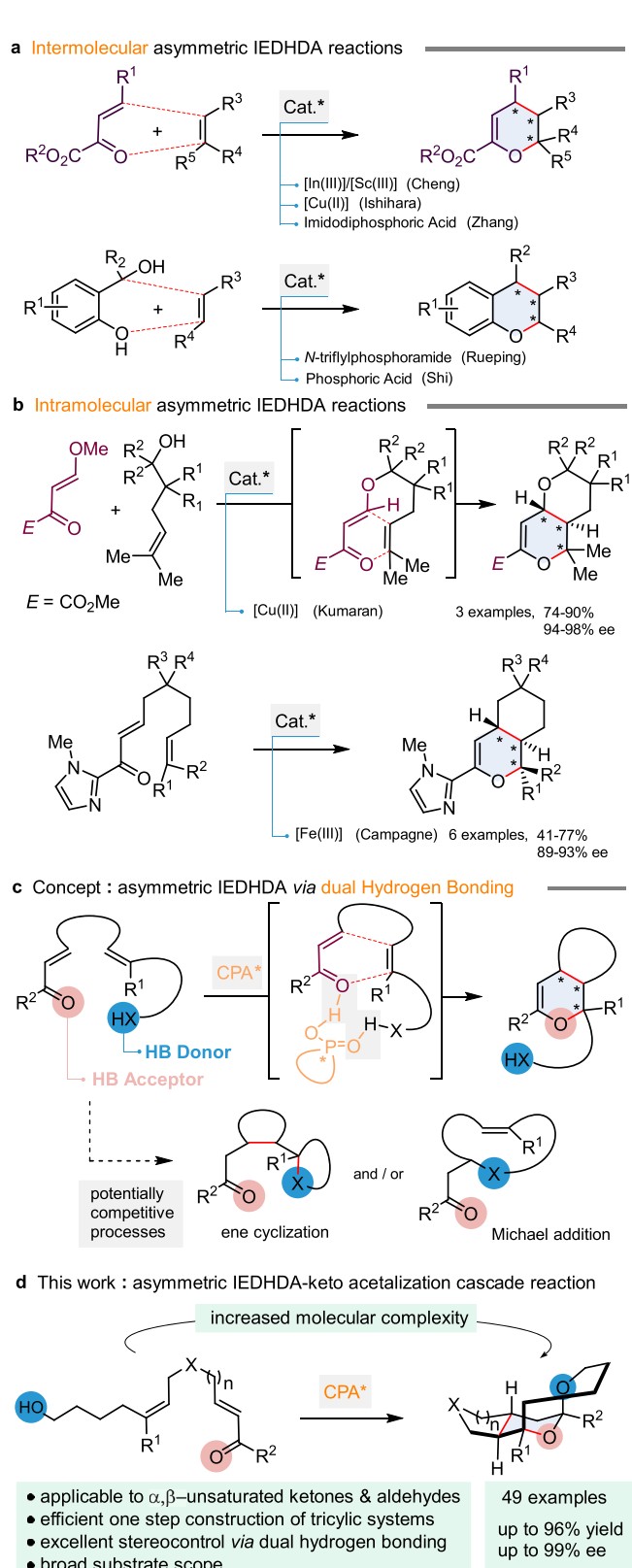

**Fig. 2 Previous studies in asymmetric IEDHDA with neutral alkene partners and our approach. a** Previous studies on intermolecular asymmetric IEDHDA reactions. **b** Previous studies on intramolecular asymmetric IEDHDA reactions. **c** Our approach to asymmetric intramolecular IEDHDA via dual hydrogen bonding interaction. **d** Access to tricyclic tetrahydropyran derivatives by chiral phosphoric acid catalyzed intramolecular IEDHDA reaction. .

hetero-Diels–Alder (HDA) reaction of β,γ-unsaturated ketoesters with allyl silanes[14]. In addition, Zhang and co-workers demonstrated that the same β,γ-unsaturated ketoester substrates could react with vinylindoles in the presence of an imidodiphosphoric acid to produce [4 + 2] cycloadducts[15]. More recently, the groups of Rueping and Shi independently reported the formation of chiral chromane derivatives by a Brønsted acid catalyzed IEDHDA reaction between a highly reactive *ortho*-quinone methide intermediate and styrene[16,17]. In comparison, catalytic and asymmetric intramolecular IEDHDA reactions of non-activated alkenes have been far less explored (Fig. 2b). An early example was reported by the group of Narasaka where an α,β-unsaturated imide reacted with a tri-substituted alkene in the presence of a chiral titanium complex[18]. However, despite its high enantioselectivity, the transformation was found inefficient. Kumaran and co-workers later demonstrated that a chiral copper complex could promote the intramolecular IEDHDA reactions between β,γ-unsaturated ketoesters and tri-substituted neutral olefins, while, very recently, Campagne and co-workers disclosed the intramolecular IEDHDA reaction of α,β-unsaturated acyl imidazole with tri-substituted neutral olefin in the presence of a Pybox/Fe(OTf)$_3$ complex. Despite being efficient and highly enantioselective, these two transformations featured a very limited scope[19,20].

To the best of our knowledge, the use of simple α,β-unsaturated ketones or aldehydes as reaction partners in catalyzed asymmetric intramolecular IEDHA with neutral alkenes has not been reported (a single example of a racemic IEDHA reaction involving an enone and vinylethylene carbonates was recently reported by Li and co-workers)[21]. The reason may be found in the low reactivity of neutral alkenes as nucleophilic component (for selected reviews on neutral alkene-participated asymmetric reactions see ref. [22], for examples on neutral alkene-participated asymmetric reactions see ref. [25])[22–31], and the difficulty to find a suitable catalyst being able to efficiently "chelate" the simple α,β-unsaturated ketone or aldehyde electrophile in order to achieve a high facial selectivity. In the synthetic approaches presented above, β,γ-unsaturated ketoesters, unsaturated oxazolidinones and acyl imidazoles were employed to efficiently complex with the metal catalyst (see Fig. 2b). In this context, we anticipated that such a challenging transformation could potentially be achieved in its intramolecular version by using a chiral Brønsted acid (Fig. 2c) (for selected reviews on the use of chiral Brønsted acids, see ref. [33])[32–37]. In the envisaged scenario, the acid would activate the α,β-unsaturated ketone or aldehyde by hydrogen bonding interaction thus favoring the nucleophilic addition of the neutral alkene (for examples of CPA activating carbonyl reactions see ref. [38])[38–41], while the enantioselectivity could be generated thanks to the presence of an additional hydrogen bond (HB) donor (HX) present in the substrate. If nucleophilic in nature, this HB donor was however considered as a potential source of side product formation since Michael addition to the α,β-unsaturated keto derivative or ene-cyclization reactions could also be possible under the reaction conditions.

In this work, we report the results of our investigations which demonstrated the feasibility of our designed approach using a chiral phosphoric acid to cyclize non-activated alkenes with α,β-unsaturated ketones or aldehydes and a remote hydroxyl group as hydrogen bond donor, to induce the enantiocontrolled formation of tricyclic tetrahydropyran derivatives by an IEDHDA—keto acetalization cascade reaction (Fig. 2d). The transformation is efficient (up to 97% yield), and exhibits excellent enantioselectivities (up to 99% ee) and a broad substrate scope (49 examples).

## Results and discussion
### Reaction optimization. 
Our studies started with model substrate (*E*)-**5a** that possesses a hydroxyl HB donor group and an α,β-

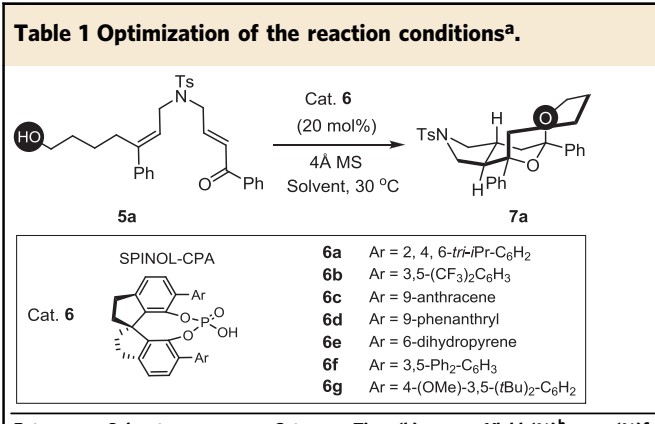

**Table 1 Optimization of the reaction conditions[a].**

| Entry | Solvent | Cat. | Time (h) | Yield (%)[b] | ee (%)[c] |
|---|---|---|---|---|---|
| 1 | CH$_2$Cl$_2$ | **6a** | 108 | 8 | 20 |
| 2 | CH$_2$Cl$_2$ | **6b** | 60 | 53 | 75 |
| 3 | CH$_2$Cl$_2$ | **6c** | 60 | 64 | 95 |
| 4 | CH$_2$Cl$_2$ | **6e** | 60 | 64 | 89 |
| 5 | CH$_2$Cl$_2$ | **6f** | 60 | 63 | 55 |
| 6 | CH$_2$Cl$_2$ | **6g** | 108 | 21 | 24 |
| 7 | CH$_2$Cl$_2$ | **6d** | 48 | 86 | 95 |
| 8 | CH$_2$Cl$_2$ | **6d** | 48 | 73 | 95[d] |
| 9 | CH$_2$Cl$_2$ | **6d** | 48 | 67 | 94[e] |
| 10 | CHCl$_3$ | **6d** | 48 | 63 | 84 |
| 11 | Toluene | **6d** | 48 | 42 | 87 |
| 12 | Cyclohexane | **6d** | 48 | 12 | 56 |
| 13 | THF | **6d** | 48 | N.R. | –[f] |
| 14 | CH$_3$CN | **6d** | 48 | 52 | 77 |

[a]Unless otherwise noted, all reactions were performed with **5a** (30 mg) and 4 Å MS (60 mg) in the presence of catalyst **6** (20 mol%) at 30 °C in 1.0 mL solvent for 48 h.
[b]Isolated yields of the single isomer formed.
[c]ee measured by chiral HPLC analysis.
[d]10 mol% of catalyst **6d** was employed.
[e]5 mol% of catalyst **6d** was employed.
[f] N.R. (no reaction).

unsaturated arylketone as the HB acceptor. A series of SPINOL-CPA were evaluated as potential catalysts to achieve the desired transformation (Table 1) (for a recent example from our laboratory using CPA see ref. [32]). The use of phosphoric acid **6a** in dichloromethane at 30 °C was rather inefficient. Most of the starting material was found unreacted after 108 h, but, gratifyingly, the formation of the desired product **7a** could be observed thus validating our approach. Notably, **7a** was produced as a single isomer and no trace of products derived from a competitive Michael addition or an ene-cyclization could be detected. The tricyclic ketal **7a** could be isolated in 8% yield and 20% ee (Table 1, entry 1). Encouraged by this result, a series of SPINOL-CPA were then screened. The enantioselectivity of the reaction could be largely improved and an optimal 95% ee could be obtained using either catalyst **6c** or **6d** (entries 3 and 7), the latest showing the highest catalytic activity (86% yield). Interestingly, when the catalytic loading was reduced to 5 mol%, the transformation remained robust affording **7a** with a lower yield after the same reaction time (67%), but with a very similar level of enantioselectivity (94% ee, entry 9). Employing other solvents in the presence of catalyst **6d** was detrimental to both the efficiency and stereoselectivity of the reaction (entries 10–14). Consequently, the use of 20 mol% of catalyst **6d** in dichloromethane at 30 °C was retained as optimal conditions to achieve the transformation with high yields and enantioselectivities, and we then investigated the scope of this IEDHDA–keto acetalization cascade reaction. Notably, the IEDHDA reaction of (*Z*)-**5a** did not proceed at all under the optimal reaction conditions.

### Scope of substrates. 
The effect of substitution on both the alkene and α,β-unsaturated ketone was surveyed. The results are shown in Fig. 3a. As for the alkene part, groups of various electronic nature were tolerated on the phenyl ring substituent R$^1$: tri-

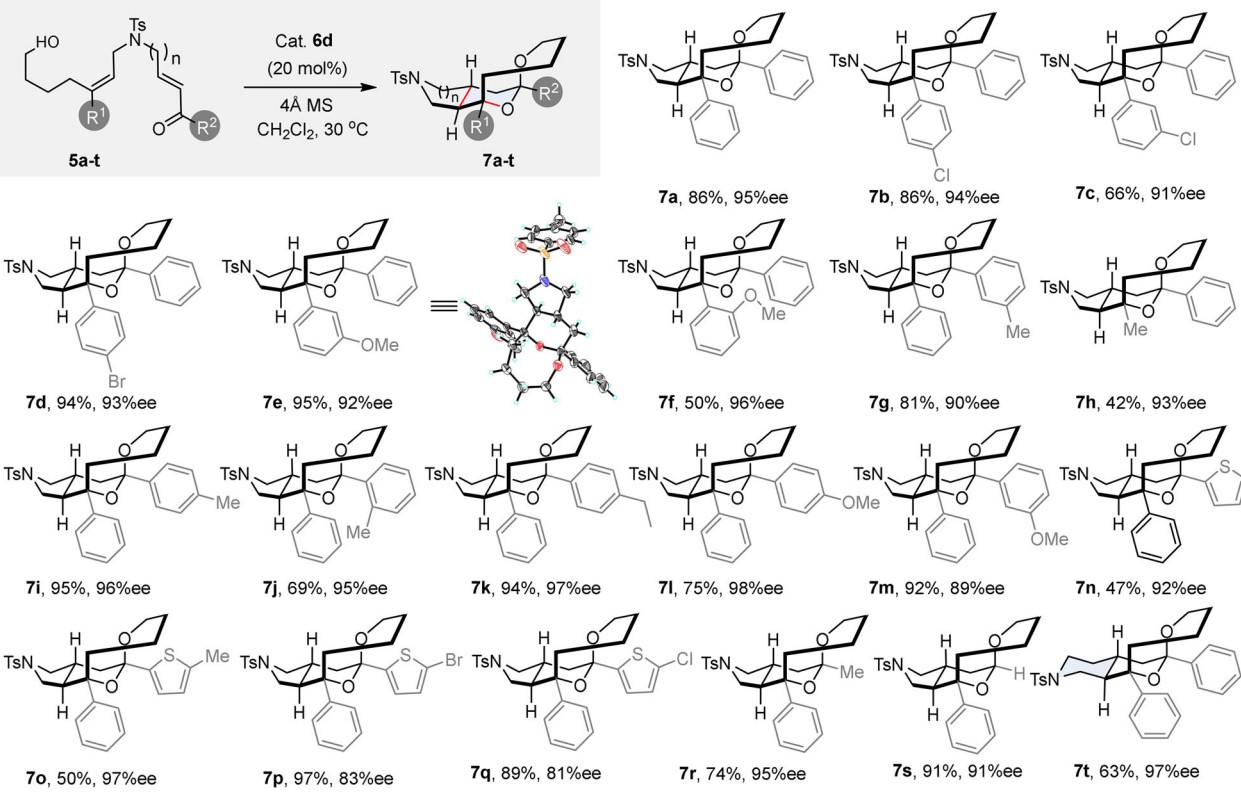

**a** Variation of the HB acceptor and alkene substituent

**7a**, 86%, 95%ee
**7b**, 86%, 94%ee
**7c**, 66%, 91%ee
**7d**, 94%, 93%ee
**7e**, 95%, 92%ee
**7f**, 50%, 96%ee
**7g**, 81%, 90%ee
**7h**, 42%, 93%ee
**7i**, 95%, 96%ee
**7j**, 69%, 95%ee
**7k**, 94%, 97%ee
**7l**, 75%, 98%ee
**7m**, 92%, 89%ee
**7n**, 47%, 92%ee
**7o**, 50%, 97%ee
**7p**, 97%, 83%ee
**7q**, 89%, 81%ee
**7r**, 74%, 95%ee
**7s**, 91%, 91%ee
**7t**, 63%, 97%ee

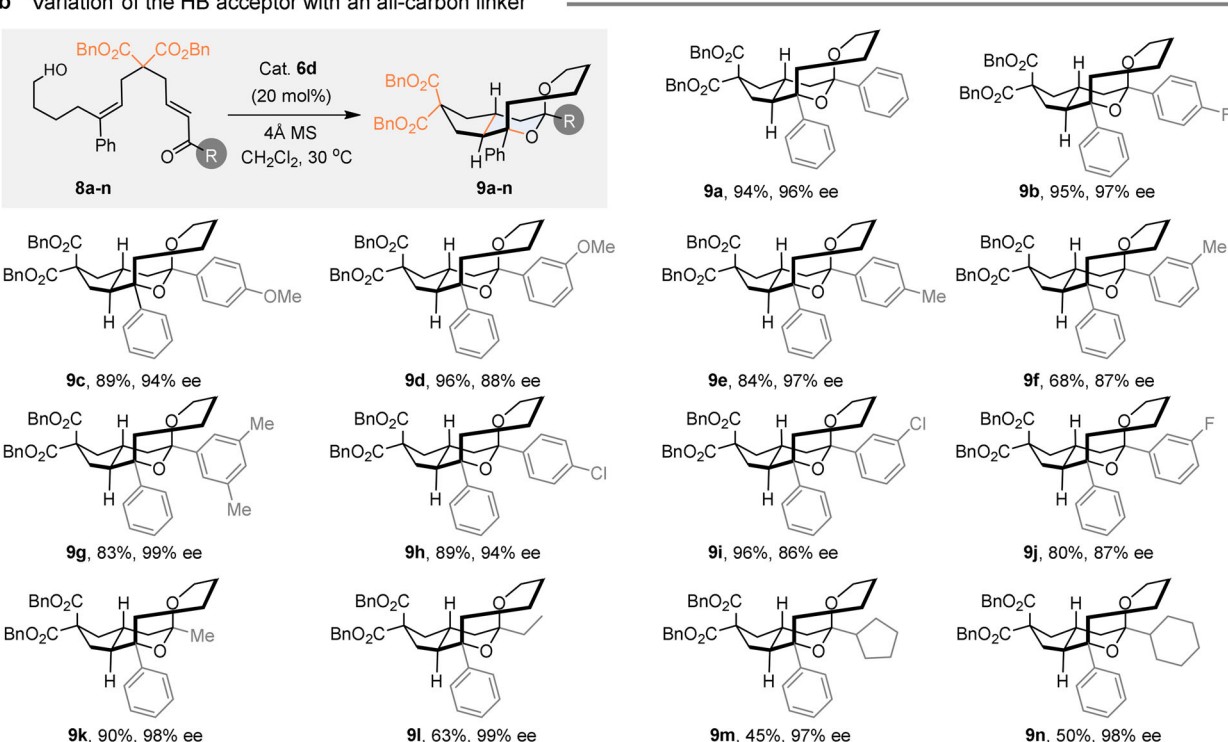

**b** Variation of the HB acceptor with an all-carbon linker

**9a**, 94%, 96% ee
**9b**, 95%, 97% ee
**9c**, 89%, 94% ee
**9d**, 96%, 88% ee
**9e**, 84%, 97% ee
**9f**, 68%, 87% ee
**9g**, 83%, 99% ee
**9h**, 89%, 94% ee
**9i**, 96%, 86% ee
**9j**, 80%, 87% ee
**9k**, 90%, 98% ee
**9l**, 63%, 99% ee
**9m**, 45%, 97% ee
**9n**, 50%, 98% ee

**Fig. 3 Scope of the reaction with α,β-unsaturated ketones as the HB acceptor. a** Variation of substituent on both the alkene (R[1]) and the α,β-unsaturated ketone (R[2]) for substrates with an tosylamide linker. **b** Variation of substituent R on the α,β-unsaturated ketone for substrates possessing an all carbon-linker. Reactions were performed with 30 mg of susbtrate **5a–t** or **8a–n** and 4 Å MS (60 mg) in the presence of catalyst **6d** (20 mol%) at 30 °C in 1.0 mL DCM for 48–60 h. Isolated yields are given; only a single stereoisomer was isolated (d.r. > 20:1); ee were measured by chiral HPLC analysis.

substituted alkene derivatives **5a–g** led to the formation of the corresponding tricyclic products **7a–g** in moderate to high yields (50–95%) and with excellent enantioselectivities (90–96% ee). The position of the substituent on the aromatic ring was shown to exert negligible influence on the facial selectivity of this reaction despite the fact that an *ortho*-MeO substituent led to reduced yield in product when compared to the analogous meta-substituted substrate (compare **7e** and **7f**). The presence of a substituent at position *ortho* may possibly alter the reactivity of the alkene partner by reducing its conjugation with the aromatic ring and/or generating unfavorable steric constraints close to the reaction centers. It is noteworthy that the transformation was not limited to the use of aryl-substituted alkenes. An alkyl group, such as a simple methyl was also compatible and for instance substrate **5h** could be converted into the IEDHDA product **7h** with an excellent ee value of 93%. The yield of the reaction was however more moderate (40%). The substitution pattern of the α,β-unsaturated ketone moiety was then investigated with substrates **5i–q**. The studies showed that in the presence of electron-rich aromatic groups, the desired tricyclic products **7i–m** could be obtained in good yields (69–95%) and with high enantioselectivities (89–98% ee). The reaction also tolerated the presence of a heterocyclic ring as exemplified by the efficient and highly selective conversion of substrates **5n–q**, that possess a substituted thiophene ring, into products **7n–q**. The reaction could also be performed with non-aromatic $R^2$ groups. An alkyl substituent such as a methyl group or a simple hydrogen atom was fully compatible: the corresponding substrates **5r** and **5s** were converted into products **7r** and **7s** with good yields (74% and 91%) and excellent enantio-control (91% ee and 95% ee). The transformation could also be extended to the formation of [6-6-8] tricyclic structures: piperidine derivative **7t** was formed for instance in 63% yield and with excellent enantioselectivity (97% ee). The relative and absolute configurations of the tricyclic ketals shown in Fig. 3a were deduced by analogy with the stereochemistry of **7e**, which was unambiguously assigned as (*R, S, R, R*) by X-ray diffraction analysis (see Supplementary Fig. 2 for more details). To further extend the diversity of products that could be produced, we also wanted to evaluate if the replacement of the NTs moiety by a carbon-based group would alter the efficiency of the process.

Gratifyingly, under the same optimal reaction condition, dibenzylmalonate derivative **8a** could be similarly converted into **9a** which was isolated in 94% yield and with 96% ee, thus demonstrating the wide applicability of the process (Fig. 3b). A series of substrate **8b–n** featuring the same malonate linker but variously substituted α,β-unsaturated ketones were then reacted (Fig. 3b). High yields and enantioselectivities (up to 96% yield and 99% ee) could be achieved with substrates **8b–j** which possess either electron-rich or poor aromatic substituents. Importantly, enones substituted by an alkyl group were suitable substrates as shown by the formation of products **9k–n**. While enantioselectivities remain uniformly high (97–99% ee) whatever the nature of the alkyl group, reduced yields were obtained in the case of cyclic alkyl substituents (**9m** and **9n**). This relative loss of efficiency was tentatively attributed to the lower stability of the ketal moiety in **9m-n**, which were observed to be more prone to hydrolysis under the acidic reaction conditions.

For such types of organocatalytic transformations, the reactivity of α,β-unsaturated aldehydes is generally quite different from that of α,β-unsaturated ketones. However, in the present case, when substrate **10a** was subjected to the optimized reaction conditions, the tricycle product **11a** could also be obtained in a good yield and with an excellent ee value (95% yield, 95% ee, Fig. 4). This reactivity could be generalized as attested by the results obtained with substrates **10b–p** bearing various malonate linkers and/or alkene substituents (Fig. 4). The nature of the

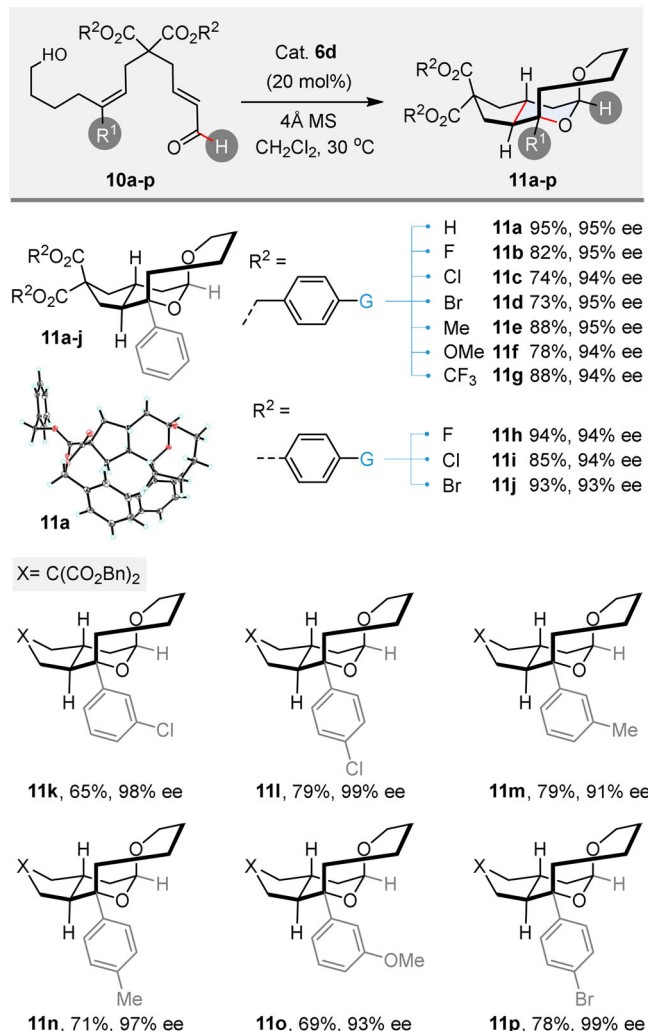

**Fig. 4 Scope of the reaction with α,β-unsaturated aldehyde as the HB acceptor.** Reactions were performed with 30 mg of substrate **10a–p** and 4 Å MS (60 mg) in the presence of catalyst **6d** (20% mol) at 30 °C in 1.0 mL DCM for 60 h. Isolated yields are given; only a single stereoisomer was isolated (d.r. > 20:1); ee were measured by chiral HPLC analysis.

malonate substituents on **10a–j** had negligible influence on the enantiocontrol (93–95% ee) and yields in tricyclic products **11a–j** were generally high (73–95%). A similar trend was observed when the nature of the aromatic substituent on the alkene was varied: slightly lower yields (65–79%) were obtained as compared to the reaction of **10a**, while the enantioselectivity remained high (91–99% ee, **11k–p**). The absolute configuration of product **11a** was unambiguously confirmed as (*R, S, R, R*) by X-ray analysis (Fig. 4, see Supplementary Fig. 3 for more details).

**Product derivatization**. To further demonstrate the synthetic usefulness of the catalytic transformation thus developed, a series of post-transformations were performed (Fig. 5). The ketal motif in **7o** could be reduced to generate the octahydropyranopyrrole **12** in 71% yield, 7:1 d.r. and without erosion of enantioselectivity (97% ee). The eight-membered cyclic ether in structure **7h** could alternatively be oxidized at room temperate to furnish lactone **13**[42]. A subsequent alcoholysis step led to chiral pyrrolidine derivative **14** that contains three contiguous stereogenic centers. The ketal moiety in product **11a** could undergo an efficient stereoselective allylation reaction to produce the bicyclic product **15**

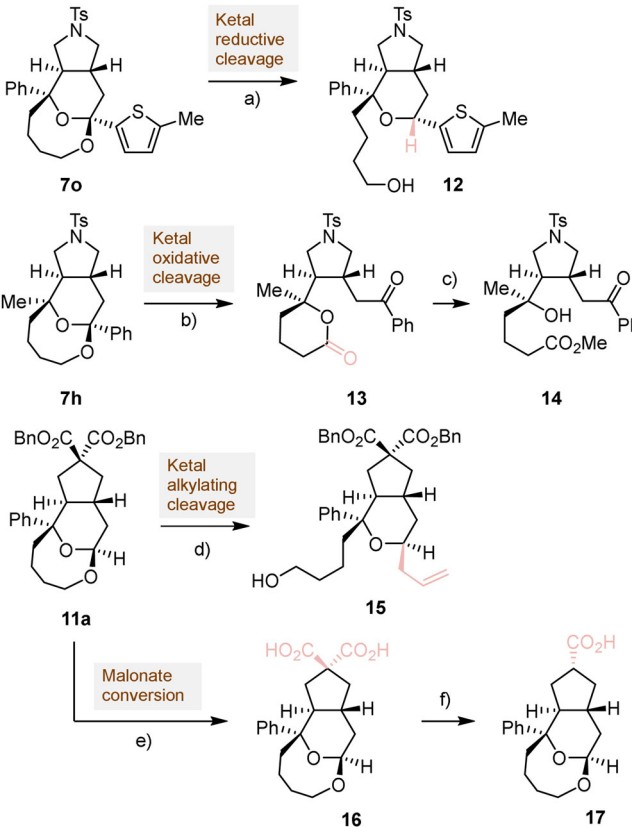

**Fig. 5 Post-transformations of IEDHDA products.** Reaction conditions: **a** DIBAL-H, CH₂Cl₂, 0 °C, 71% (d.r. = 7:1), 97%ee. **b** RuCl₃, NaIO₄, CH₃CN/CHCl₃/H₂O, RT, 45%, 93%ee. **c** K₂CO₃, MeOH, RT, 65%, 93%ee. **d** Allyltrimethylsilane, SnCl₄, −30 °C to RT, CH₂Cl₂, 68%, 95%ee. **e** Pd/H₂, MeOH, quant.; **f** Pyridine, reflux, 78%, 95%ee.

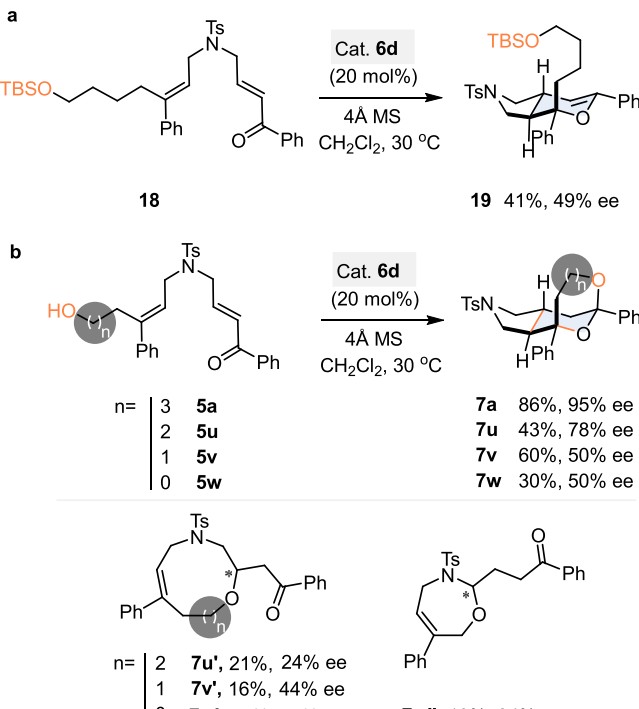

**Fig. 6 Probing the HB interaction. a** IEDHDA reaction with a substrate lacking the HB donor. **b** Effect of the HB donor-alkene linker size on the enantiocontrol.

in 68% yield. Finally, it was shown that the same tricyclic compound **11a** could be hydrogenated to quantitatively afford the dicarboxylic acid **16**, which in turn could be transformed into monoacid **17** with the generation of a new stereogenic center. This selection of classical functional group conversions demonstrates the synthetic potential of the present asymmetric catalytic IEDHDA reaction to generate structural diversity.

**Mechanistic studies.** We then worked at determining the mechanism of the reaction and rationalizing the observed high enantioinduction. We first studied the HB donor role of the hydroxyl group in the process (Fig. 6a). To this purpose, the TBS-protected substrate **18**, derivative from model substrate **5a**, was prepared and subjected to the optimal reaction conditions. The IEDHDA dihydropyran product **19** was formed in a moderate 41% yield that was attributed to the tendency of **19** to be hydrolyzed under the acidic reaction conditions as well as during the purification step by flash column chromatography. Most importantly, **19** was produced with a significantly reduced 49% ee, as compared to the formation compound **7a** from the parent free OH substrate **5a** (86% yield, 95% ee, see Table 1, entry 7). This result suggests that the hydroxyl group on **5a** may function not only as the nucleophile in the cascade process leading to the formation of the ketal product, but also as a remote hydrogen donor to enhance the interaction between substrate **5a** and catalyst **6d**, and thus improving the facial selectivity in the process. If catalyst **6d** allows the IEDHDA reaction to take place through a dual hydrogen bonding interaction, the strength of the interaction between the substrate and the catalyst and the consequent spatial organization should logically be affected by the distance between the hydroxyl moiety and carbonyl group.

Based on this speculation, a series of substrate **5u–w** possessing alkene/hydroxyl linkers of various sizes were synthesized and reacted under the standard reaction conditions. The IEDHDA reactions with **5u–w** took place and could provide the corresponding tricyclic products **7u-w** as well as the by-products **7u′–7w′** due to the competitively intramolecular Michael addition of the hydroxyl group to the enone moiety and side product *N, O*-acetal **7w″** in the case of **5w** as well. The results obtained (Fig. 6b) clearly show a decrease of the ee values when the linker is shortened. Substrates **5v** and **5w** afforded the desired tricyclic product **7v** and **7w** but in moderate yield (respectively, 60% and 30% yield) and both with a 50% ee which appeared to be the same enantiocontrol obtained for the conversion of TBS-ether **18** into **19**. These results suggest that a dual hydrogen bonding interaction may not be in play for the conversion of **5v** and **5w** due to an unfavorable spatial distance between the hydroxyl HB donor and the carbonyl HB acceptor. In these cases, the lack of structural flexibility in the substrates may lead to the inability of catalyst **6d** to properly "coordinate" to **5v** or **5w** in a dual manner.

On the basis of these experiments and by taking into account the absolute configuration of the tricyclic products obtained, we proposed the catalytic mechanism as shown in Fig. 7. In this model, the terminal hydroxyl group in **5a** would interact with the P = O functionality of catalyst **6d**, while the P−OH moiety would electrophilically activate the carbonyl group in **5a**. Both HB interactions between **5a** and **6d** should be synergistically involved to induce the IEDHDA reaction via a spatially well-organized transition state such as **TS1-1**. The resulting dihydropyran intermediate **20** would be further activated by the acidic catalyst to promote the ketalization step leading to **7a** via transition state **TS2**. From a selectivity point of view, it is proposed that the *Si* face of the trisubstituted alkene could be preferentially reacted with the *Si* face of the conjugated enone (as shown in **TS1-1**).

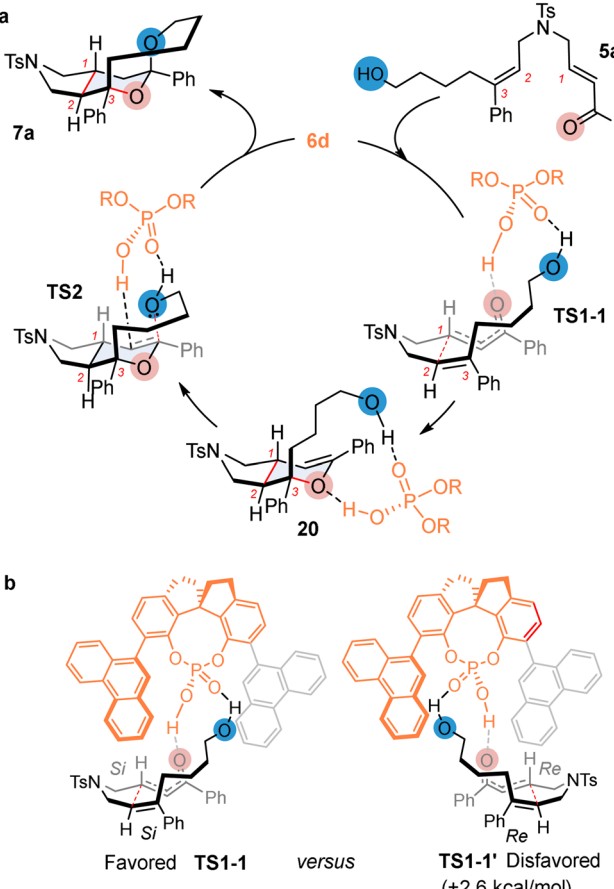

**Fig. 7 Mechanistic proposal. a** Proposed catalytic cycle. **b** Transition state model of stereocontrol.

This mechanism was supported by density-functional theory (DFT) calculations performed on substrate **5a** (see Supplementary Discussion for more details). The computational studies showed **TS1-1** to be energetically more favorable than **TS1-1′** by 2.6 kcal/mol, a value in agreement with the level of enantioselectivity experimentally observed. Interestingly, the formation of the dihydropyran moiety was calculated to proceed in a stepwise manner. The cyclization process would be initiated by the formation of the $C_1$–$C_2$ bond via **TS1-1**, a step that would be rate determining. A subsequent facile $C_3$–O bond formation (1.5 kcal/mol) would deliver intermediate **20**.

In conclusion, we have developed a chiral phosphoric acid-promoted enantioselective IEDHDA-keto-acetalization cascade transformation that involves the initial reaction of either α,β-unsaturated ketones or aldehydes with non-activated alkenes to ultimately produce trans-fused octahydropyranopyrrole and trans-fused tetrahydropyrano-cyclopentane tricyclic frameworks. The transformation was shown highly enantioselective and could be applied to a variety of substrates. The design of the remote dual HB interaction was key to achieving high chiral induction in this challenging transformation. Application of this concept to the development of other organocatalytic reactions is currently underway in our laboratory.

## Methods

**General procedures for the IEDHDA reactions**. Substrate **5a–t**, **8a–n** or **10a–p** (30 mg, 1.0 eq.) was dissolved in anhydrous DCM (1.0 ml), then 4 Å molecular sieves (60.0 mg) and catalyst **6d** (0.2 eq.) were added. The reaction mixture was sealed and stirred at 30 °C and monitored by TLC (generally for 48–60 h). When the reaction was completed, the mixture was directly purified by silica gel column to afford products **7a–t**, **9a–n**, or **11a–p**, respectively.

## Data availability

All data generated in this study are provided in the Supplementary Information/Source Data file. The X-ray crystallographic coordinates data used in this study are available in the Cambridge Crystallographic Data Center (CCDC) under accession code CDCC-2069300 (**7e**) and CDCC-2069301 (**11a**) [www.ccdc.cam.ac.uk/data_request/cif]. Source data are provided with this paper.

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

## Acknowledgements

Financial support for this work was provide by the National Natural Science Foundation of China (Grant No. 21772207, L.-L.W), CAS Light of West China Program (No. Y8252211W1, L.-L.W), Youth Innovation Promotion Association CAS (No. Y9244251T1, L.-L.W) and "High-Level Talent Program" of Yunnan Province (No. Y93D321261, L.-L.W). F.G. thanks the University of Ottawa and the Natural Sciences and Engineering Research Council for support.

## Author contributions

L.-L.W. and Z.Z. conceived of and directed the project, and F.G. co-wrote the manuscript. M.J. and C.T. discovered and developed the asymmetric reactions, and co-wrote the Supplementary Information. Y.L., S.Y., Y.-T.Y., L.P., X.-N.L., W.Z. performed part of synthetic experiments, and W.Z. performed the DFT calculation.

## Competing interests

The authors declare no competing interests.
