## [Peer Review File · Nature Communications]

REVIEWER COMMENTS

Reviewer #1 (Remarks to the Author):

In this manuscript, Zuo and Wang and coworkers reported an interesting enantioselective intramolecular inverse electron demand hetero Diels-Alder reaction with excellent results. An organocatalytic systems has been optimized for the reaction of rich alkenes and α,β -unsaturated ketones and aldehydes.

The introduction is well written and was easy to follow when the methodology has been presented considering the state of the art. However, at first the introduction was focussed on the importance of trans-fused [5,6] bicyclic ring cores. Even though I would have mentioned them I am not sure if I would have presented and discuss so many structures when they are not reachable by the presented methodology.

The reaction scope is very broad and the reaction performance shown very high.

Thus, I have only a couple of curiosities:

Are aliphatic ketones suitable substrates for the cyclization to give rise the nitrogen based heterocycles?

I would like to see if alpha-beta-unsaturated aldehydes are suitable for the synthesis of nitrogen-based heterocycles like in the first substrate scope (Figure 3A).

Regarding the derivatizations I would only point out that I would find interesting to see (in the scheme) if the enantiopurity of the new compounds prepared suffers erosion in their enantiomeric excesses.

In my opinion, the presented manuscript represents a very impressive reaction that present excellent results although I believe that a 20 mol% of catalyst loading can be considered a bit too high for the kind of catalysis presented, taking into account the high molecular weight of the phosphoric acid employed. Although I have doubts about if the presented methodology represents a too specific reaction considering the multidisciplinary of the journal, I think that I would recommend this borderline case to be published in Nature Communications after addressing the above-mentioned comments.

Reviewer #2 (Remarks to the Author):

This manuscript describes enantioselective intramolecular inverse-electron-demand-hetero-Diels-Alder reaction catalyzed by chiral SPINOL-derived phosphoric acid. In this reaction, 4-hydroxybutyl group of substrates was important to induce high enantioselectivity. The importance of a hydrogen bonding interaction between 4-hydroxybutyl group of the substrate and phosphoryl oxygen of the catalyst is considered. However, experimental evidences are not enough for the proposed mechanism shown in Figure 7. Synthetically, requirement of the 4-hydroxybutyl group for high enantioselectivity is negative because the 4-hydroxybutyl group cannot remove from products. Natural products shown in Figure 1 cannot be synthesized by using this reaction. These results are excellent, but the novelty is not enough as a paper in Nat. Commun.

Reviewer #3 (Remarks to the Author):

This manuscript describes dual-hydrogen bonding assisted intramolecular enantioselective inverse-electron-demand hetero Diels-Alder reaction of omega-alkenyl alpha, beta-unsaturated carbonyl compounds having long side chain bearing hydroxy group, affording tetrahydropyrane fused with five-membered ring with trans juncture featuring intramolecular ketal. The idea of dual-hydrogenbonding activation would intrigue interest of general reader. The chemical yields are generally high as well as enantioselectivities and scope of the reaction is wide. Accordingly, the reviewer thinks that the manuscript can be accepted for a publication in this journal after minor revision listed below.

1) Page 3, the right column, line 12: The reason of the difference in chemical yield between 13e and 13f should be explained. For example, the ortho-methoxy group interferes efficient conjugation, hence the electron donating effect of the alkene is reduced.

2) Even O-protected substrate 24 gave the intramolecular cycloadduct 25 although the chemical yield and enantioselectivity were low (page 6, the left column, Fig. 6). Why the reaction of (Z)-11a, the geometrical isomer of 11a, did not proceed at all (page 6, ref. 45).

3) Page 6, the right column, Fig. 7: For the formation of intramolecular ketal formation from the initial cycloadduct 26, the authors proposed protonation from top-face of 26 (TS2 in Fig 7).

However, top-face protonation should give unstable twist-boat form whereas bottom-face protonation should yield the more stable chair form cation. Accordingly, the product 13a should be produced bottom-face protonation.

4) There are many typological errors in References.

Ref.4: The title should be given.

Ref. 10: The full authors' names should be given.

Ref. 13: The space between the comma and beta should be deleted. The space between the comma and 4 should be deleted. 2018 should be typed in plane.

Ref. 17: The hyphen between stereose and lectivity should be deleted. One space just before Olefins should be deleted.

Ref. 19: H of 2H should be typed in italic form.

Ref. 23: The hyphen between antio and selective should be deleted.

Ref. 24: The full authors' names should be given.

Ref. 25: The hyphen between car and bonates should be deleted. The full authors' names should be given.

Ref. 32: The hyphen between oycliza and tion should be deleted.

Ref. 33: The full authors' names should be given.

Ref. 36: The journal name should be typed in italic form.

Ref: 45: 2017 should be typed in plain form.

Thank the referees very much for reviewing our manuscript and for their critical comments as well as some good suggestions which are helpful to further improve the quality of this manuscript. We have revised our manuscript based on all the reviewers' comments. Please see the revised Manuscript with the highlighted changes.

The following answers are point-by-point responses to the reviewers.

Reviewer 1:

Comments:

In this manuscript, Zuo and Wang and coworkers reported an interesting enantioselective intramolecular inverse electron demand hetero Diels-Alder reaction with excellent results. An organocatalytic systems has been optimized for the reaction of rich alkenes and a,b-unsaturated ketones and aldehydes.

The introduction is well written and was easy to follow when the methodology has been presented considering the state of the art. However, at first the introduction was focussed on the importance of trans-fused [5,6] bicycles ring cores. Even though I would have mentioned them I am not sure if I would have presented and discuss so many structures when they are not reachable by the presented methodology.

Answer: We thank the reviewer for this kind and valuable comments. Following the suggestion/advice, the first part of the introduction section has been shortened and the number of structures presented in Figure 1 has been limited to 4. These selected molecules are representative natural products or bioactive molecules featuring either a *trans*-fused tetrahydropyranocyclopentane motif (substructure A in Figure 1) or an octahydropyranopyrrole motif (substructure B in Figure 1). This change would make the introduction more concise while mentioning the interest of the present method. Additionally, the synthetic method that has been developed was not primarily aimed to be applied in total synthesis, but more likely to access structural fragments that are of importance (given the number of bioactive compounds featuring either substructure A or B that have been reported in the literature). As exemplified by the large scope done, the method would be an efficient platform to access a variety of derivatives possessing these structural units.

The reaction scope is very broad and the reaction performance shown very high.

Thus, I have only a couple of curiosities:

Are aliphatic ketones suitable substrates for the cyclization to give rise the nitrogen based heterocycles? I would like to see if alpha-beta-unsaturated aldehydes are suitable for the synthesis of nitrogen-based heterocycles like in the first substrate scope (Figure 3A).

Answer: Following the suggestion, substrate **5r** that possesses a methyl ketone moiety has been synthesized and subjected to the optimized reaction conditions. The

corresponding cyclized product **7r** was formed in good yield (74%) and with an excellent enantioselectivity (95%ee). In addition, the analogous α,β -unsaturated aldehyde **5s** reacted efficiently and selectively to product **7s** (91% yield, 91%ee). These two additional examples, which have been included in the revised manuscript in Figure 3a, suggest that the method is not limited to the use of aromatic substituted ketones but also to aliphatic ones and simple aldehydes.

Regarding the derivatizations I would only point out that I would find interesting to see (in the scheme) if the enantiopurity of the new compounds prepared suffers erosion in their enantiomeric excesses.

Answer: Following this question, the corresponding racemic products have been synthesized and the ee values of the derivatization products **12-17** (see Figure 5 of the revised manuscript) have been measured. These values are now shown in the notes associated with Figure 5. Notably, it was shown that the enantiopurity of these compounds did not suffer erosion during the derivatization processes.

In my opinion, the presented manuscript represents a very impressive reaction that present excellent results although I believe that a 20 mol% of catalyst loading can be considered a bit too high for the kind of catalysis presented, taking into account the high molecular weight of the phosphoric acid employed.

Answer: We thank the reviewer for this kind and encouraging comment. The 20 mol% catalyst loading was used during this work to prove the concept of reactivity, perform the scope of the reaction and demonstrate the large applicability of the method. We agree with the referee that such a loading may be important, therefore we have focused our attention during this manuscript revision on the possibility to use a lower catalyst loading to achieve the same level of reaction efficiency. The reaction of model substrate **5a** was conducted with either 10 mol% and 5 mol% of the catalyst under the standard experimental conditions (in DCM at 30 °C in the presence of 4Å MS for 48 hours) and delivered the desired product **7a** in good yields and excellent selectivities (for 10 mol%: 73% yield, 95%ee; for 5 mol%: 67% yield, 94% ee). These results, which have been added in entries 8 & 9 of Table 1 in the revised manuscript, demonstrate that a lower amount of catalyst could be employed for the reaction and that the reduced loading only (slightly) affected the yield for the same the reaction time.

Although I have doubts about if the presented methodology represents a too specific reaction considering the multidisciplinary of the journal, I think that I would recommend this borderline case to be published in Nature Communications after addressing the above-mentioned comments.

Answer: We highly appreciate the positive comments from the referee. Beyond the synthetic transformation, the present work described the successful application of a new concept *i.e.* chiral induction by dual hydrogen bonding interaction between a reactive acceptor and a remote non-reactive HB donor. This general process could be implemented to other transformations than the present IEDHDA.

Reviewer: 2

Comments:

This manuscript describes enantioselective intramolecular inverse-electron-demand-hetero-Diels–Alder reaction catalyzed by chiral SPINOL-derived phosphoric acid. In this reaction, 4-hydroxybutyl group of substrates was important to induce high enantioselectivity. The importance of a hydrogen bonding interaction between 4-hydroxybutyl group of the substrate and phosphoryl oxygen of the catalyst is considered. However, experimental evidences are not enough for the proposed mechanism shown in Figure 7.

Answer: The experimental mechanistic studies, whose results are shown in Figure 6 in the revised manuscript, clearly support a process in which a remote hydrogen bonding interaction with the free alcohol is key to achieve high enantioinduction. When the hydroxyl HB donor was replaced by a TBS-protected ether a very significant drop in ee was observed (49% ee versus 95% ee) and the yield was also moderate. The length of the tether between the hydroxyl HB donor and the alkene was also shown to alter the enantioselectivity. To support the mechanistic proposal shown in Figure 7, additional DFT studies have been performed. A summary of these investigations has been inserted at the end of the manuscript section entitled “mechanistic studies” and the full details can be found in the Supplementary information (Supplementary Figure 3).

Results of the DFT calculations support the mechanism proposed in Figure 7. Key results are as follow: 1) From model substrate **5a**, the proposed transition state **TS1-1** that involves the 2 key HB interactions, was found to be energetically more favorable than **TS1-1'** by 2.6 kcal/mol, a value in agreement with the level of enantioselectivity experimentally observed for the production of **7a**; 2) the formation of the dihydropyran moiety was calculated to proceed in a stepwise manner. The cyclization process would

be initiated by the formation of the C₁-C₂ bond via **TS1-1**, a step that would be rate determining (15 kcal/mol). A subsequent facile C₃-O bond formation (1.5 kcal/mol) would deliver intermediate **20**; 3) activation of the enol moiety in **20** by the catalyst, would produce an oxonium intermediate that would rapidly collapse into ketal **7a**. The activation energy barrier for the conversion of **20** to the final ketal was calculated to be 10.4 kcal/mol.

Synthetically, requirement of the 4-hydroxybutyl group for high enantioselectivity is negative because the 4-hydroxybutyl group cannot remove from products.

Answer: The 4-hydroxybutyl moiety is indeed a structural requirement to proceed with the dual HB interactions necessary to achieve high enantioinduction during the cyclization step. We understand and agree with the reviewer that this motif would not be easily removed from the product if required. However, we would like to stress that the prime goal of this study was to highlight the possibility to perform such a level of enantioinduction following the new concept of dual HB interaction with a remote HB donor. This, we believe, has been achieved as attested by the large reaction scope and the high level of enantioselectivity obtained. As shown in the derivatization work we performed (Figure 5), the ketal moiety could be cleaved under reductive or oxidative conditions to release the hydroxybutyl chain (products **12&15**) or to produce carboxylic acid derivatives (products **13&14**). With the idea to produce a large variety of compounds possessing either a *trans*-fused tetrahydropyranocyclopentane motif (substructure A in Figure 1) or an octahydropyranopyrrole motif (substructure B in Figure 1), compounds **12-15** could be useful building blocks for further derivatization (ester, amides ...). In order to initiate demonstrating that the hydroxybutyl group that was initially chosen to validate our concept could be modified while allowing the reaction to proceed, the following transformation with a simple glycol derivative has been performed under standard experimental conditions. Indeed, it has been previously reported in the literature that ethylene glycol derivatives could be cleaved under oxidative and/or acidic conditions (see for examples: *Org. Lett.* **2012**, *14*, 3218; *Tetrahedron* **2006**, *62*, 7056; *J. Chem. Pharm. Res.* **2014**, *6*, 47). While the yield and enantioselectivity were not perfect (53%, 60% ee), this very preliminary result indicates the potential to develop a cleavable linker for this reaction, what would be part of our future investigations in this field.

Natural products shown in Figure 1 cannot be synthesized by using this reaction.

Answer: As previously stated in our reply to reviewer 1, the synthetic method that has been developed was not primarily aimed to be applied in total synthesis, but more likely

to access structural fragments that are of importance (given the number of bioactive compounds featuring either substructure A - *trans*-fused tetrahydropyranocyclopentane motif - or B - octahydropyranopyrrole motif - that have been reported in the literature, see Figure 1). As exemplified by the large scope, the method would be an efficient platform to access a variety of derivatives possessing these structural units. To prevent any misunderstanding for the reader, the introduction has been shortened and only 4 representative examples of natural products and/or bioactive molecules have been kept in our revised manuscript.

These results are excellent, but the novelty is not enough as a paper in Nat. Commun.

Answer: Also thank the reviewer's critical comment, which is helpful to our research work in the future.

Actually, neutral alkenes have been longstanding problematic substrates for asymmetric catalysis, due to their low (nucleophilic) reactivity and the consequent difficulty to efficiently control the enantioselectivity of the reactions in which they could be involved in. As for asymmetric IEDHDA type reactions, neutral alkenes have been, for instance, largely underconsidered and their reaction with α , β -unsaturated ketones or aldehydes has remained very challenging (no example had been reported so far in the literature). Practically, our work not only demonstrates that neutral alkenes and α , β -unsaturated ketone or aldehyde can be decent partners, but it also proves that their reaction can be performed with good yields, high enantioselectivity levels and with a very broad applicability (40+ examples). In addition, such a reactivity and selectivity could be achieved thanks to a new concept of dual HB interaction with an extra HB donor (that generates a weak interaction), that allows the preorganization of the substrate to efficiently control the reaction. This concept, that was inspired by enzymatic catalysis for which multiple HB interactions are concomitantly involved, could in principle be applied to other challenging asymmetric reactions, and we therefore believe this work would attract the interest of the general reader. We hope that, the additional studies performed during the revision process including more especially the applicability of the process to cleavable linkers containing substrates and the DFT calculations that support our mechanistic proposal, would convince the present reviewer of the value and the novelty of our work.

Reviewer: 3

Comments:

This manuscript describes dual-hydrogen bonding assisted intramolecular enantioselective inverse-electron-demand hetero Diels-Alder reaction of omega-alkenyl α , β -unsaturated carbonyl compounds having long side chain bearing hydroxy group, affording tetrahydropyran fused with five-membered ring with *trans* juncture featuring intramolecular ketal. The idea of dual-hydrogen bonding activation would intrigue interest of general reader. The chemical yields are generally high as well as enantioselectivities and scope of the reaction is wide. Accordingly, the reviewer

thinks that the manuscript can be accepted for a publication in this journal after minor revision listed below.

Answer: We appreciate the reviewer for this kind and positive comments on the quality of our work.

1) Page 3, the right column, line 12: The reason of the difference in chemical yield between 13e and 13f should be explained. For example, the ortho-methoxy group interferes efficient conjugation, hence the electron donating effect of the alkene is reduced.

Answer: We agree an explanation on this variation of reactivity should be given. According to the reviewer's suggestion, the following comment has been inserted into the revised manuscript: "The presence of a substituent at position ortho may possibly alter the reactivity of the alkene partner by reducing its conjugation with the aromatic ring and/or generating unfavorable steric constraints close to the reaction centers."

2) Even O-protected substrate 24 gave the intramolecular cycloadduct 25 although the chemical yield and enantioselectivity were low (page 6, the left column, Fig. 6). Why the reaction of (Z)-11a, the geometrical isomer of 11a, did not proceed at all (page 6, ref. 45).

Answer: We appreciate this comment on the lack of reactivity. To attempt rationalizing the non-reactivity of substrate (Z)-5a in the revised manuscript Ref. 42 (previously numbered (Z)-11a), compounds (Z)-18 and geometrical isomer of 5h with a methyl substituent on the alkene were synthesized and reacted under the optimized reaction conditions. In both cases, no reaction could be observed. The change in the configuration of the alkene seems to play a critical role in the reaction. While the extra HB donor was present in the following substrate geometrical isomer of 5h, no reaction could be observed as well. The lack of reactivity may be the result of a negative interaction at the transition state between the phenyl (or methyl) on the (Z)-alkene and the catalyst activating the enone moiety.

3) Page 6, the right column, Fig. 7: For the formation of intramolecular ketal formation from the initial cycloadduct 26, the authors proposed protonation from top-face of 26 (TS2 in Fig 7). However, top-face protonation should give unstable twist-boat form whereas bottom-face protonation should yield the more stable chair form cation. Accordingly, the product 13a should be produced bottom-face protonation.

Answer: We thank the reviewer for this constructive comment. The route presented in

Figure 7 was supported by the DFT calculations studies (see Supplementary Figure 3 in Supplementary information). Even if the formation of the oxocarbenium by top-face protonation of the enol moiety would *a priori* appear as less likely, we believe that in the present case, the energy gain *via* H bonding between the alcohol and the catalyst – making the alcohol more nucleophilic - and *via* preorganization would make the process viable. While the suggested activation of the enol from the bottom face cannot be ruled out, DFT calculations showing the catalyst being bound to the substrate during the whole process, suggest that a decoordination/reactivation process would most probably be more energy demanding.

4) There are many typographical errors in References.

Ref. 4: The title should be given.

Answer: Thanks the reviewer for the very careful correction. The title has been added in the revised manuscript (Ref. 2).

Ref. 10: The full authors' names should be given.

Answer: Thanks for pointing out these omissions. Full authors' names have been added in the revised manuscript (Ref. 6).

Ref. 13: The space between the comma and beta should be deleted. The space between the comma and 4 should be deleted. 2018 should be typed in plane.

Answer: Thanks for pointing out these formatting errors. Corrections have been made accordingly in the revised manuscript (Ref. 9).

Ref. 17: The hyphen between stereose and lectivity should be deleted. One space just before Olefins should be deleted.

Answer: Thanks for pointing out these formatting errors. Corrections have been made accordingly in the revised manuscript (Ref. 13).

Ref. 19: H of 2H should be typed in italic from.

Answer: Thanks for pointing out these formatting errors. Corrections have been made accordingly in the revised manuscript (Ref. 15).

Ref. 23: The hyphen between antio and selective should be deleted.

Answer: Thanks for pointing out these formatting errors. Corrections have been made accordingly in the revised manuscript (Ref. 19).

Ref. 24: The full authors' names should be given.

Answer: Thanks for pointing out these omissions. Full authors' names have been added in the revised manuscript in the revised manuscript (Ref. 20).

Ref. 25: The hyphen between car and bonates should be deleted. The full authors' names should be given.

Answer: Thanks for pointing out these formatting errors and omissions. Corrections have been made and full authors' names have been added in the revised manuscript (Ref. 21).

Ref. 32: The hyphen between ocycliza and tion should be deleted.

Answer: Thanks for pointing out these formatting errors. Corrections have been made accordingly in the revised manuscript (Ref. 28).

Ref. 33: The full authors' names should be given.

Answer: Thanks for pointing out these omissions. Full authors' names have been added in the revised manuscript (Ref. 29).

Ref. 36: The journal name should be typed in italic form.

Answer: Thanks for pointing out this formatting error. Correction has been made accordingly in the revised manuscript (Ref. 32).

Ref: 45: 2017 should be typed in plain form.

Answer: Thanks for pointing out this formatting error. Correction has been made accordingly in the revised manuscript (Ref. 41).

REVIEWERS' COMMENTS

Reviewer #1 (Remarks to the Author):

Considering the modifications showed by the authors (not only following my advices but valuable suggestions made by the other two reviewers) in the new version of the manuscript, I would recommend its publication in Nature Commun

Reviewer #3 (Remarks to the Author):

The reviewer carefully read the revised manuscript and found the manuscript reasonably revised. The introduction was made concise and the mechanism supported DFT calculation is rational. Accordingly, the reviewer agree that this manuscript will be accepted in this journal.

Thank the referees very much for reviewing our manuscript again and for their comments. We have revised our manuscript based on all the reviewers' comments and editorial requests. Please see the revised Manuscript with the highlighted changes.

The following answers are point-by-point responses to the reviewers.

Reviewer #1:

Considering the modifications showed by the authors (not only following my advices but valuable suggestions made by the other two reviewers) in the new version of the manuscript, I would recommend its publication in Nature Commun

Answer: We thank the reviewer for reviewing our manuscript again and for this positive comments on the revised manuscript.

Reviewer #3:

The reviewer carefully read the revised manuscript and found the manuscript reasonably revised. The introduction was made concise and the mechanism supported DFT calculation is rational. Accordingly, the reviewer agree that this manuscript will be accepted in this journal.

Answer: We highly appreciate the reviewer for his carefully reading on our revised manuscript and for this positive comments on the revised manuscript.

Thanks again for all referees on reviewing our manuscript.